# Structure-Function Studies of Polymyxin B Lipononapeptides

**DOI:** 10.3390/molecules24030553

**Published:** 2019-02-02

**Authors:** Alejandra Gallardo-Godoy, Karl A. Hansford, Craig Muldoon, Bernd Becker, Alysha G. Elliott, Johnny X. Huang, Ruby Pelingon, Mark S. Butler, Mark A. T. Blaskovich, Matthew A. Cooper

**Affiliations:** Institute for Molecular Bioscience, The University of Queensland, Brisbane, Queensland 4072, Australia; a.gallardogodoy@uq.edu.au (A.G.-G.); cmouldy@yahoo.com.au (C.M.); bbeckerr@gmail.com (B.B.); a.elliott@imb.uq.edu.au (A.G.E.); johnny.xiao.huang@gmail.com (J.X.H.); r.pelingon@imb.uq.edu.au (R.P.); m.butler5@uq.edu.au (M.S.B.)

**Keywords:** polymyxin, antibiotic resistance, antibiotics, nonapeptide, lipopeptide

## Abstract

The emerging threat of infections caused by highly drug-resistant bacteria has prompted a resurgence in the use of the lipodecapeptide antibiotics polymyxin B and colistin as last resort therapies. Given the emergence of resistance to these drugs, there has also been a renewed interest in the development of next generation polymyxins with improved therapeutic indices and spectra of action. We report structure-activity studies of 36 polymyxin lipononapeptides structurally characterised by an exocyclic FA-Thr^2^-Dab^3^ lipodipeptide motif instead of the native FA-Dab^1^-Thr^2^-Dab^3^ tripeptide motif found in polymyxin B, removing one of the positively charged residues believed to contribute to nephrotoxicity. The compounds were prepared by solid phase synthesis using an on-resin cyclisation approach, varying the fatty acid and the residues at position 2 (P2), P3 and P4, then assessing antimicrobial potency against a panel of Gram-negative bacteria, including polymyxin-resistant strains. Pairwise comparison of *N*-acyl nonapeptide and decapeptide analogues possessing different fatty acids demonstrated that antimicrobial potency is strongly influenced by the *N*-terminal L-Dab-1 residue, contingent upon the fatty acid. This study highlights that antimicrobial potency may be retained upon truncation of the *N*-terminal L-Dab-1 residue of the native exocyclic lipotripeptide motif found in polymyxin B. The strategy may aid in the design of next generation polymyxins.

## 1. Introduction

The polymyxins (Pmx) are natural product polycationic lipodecapeptides produced by *Paenibacillus polymyxa* (Figure 1), exemplified by polymyxin B **1** (PmxB) and E **2** (also known as colistin) [1,2,3,4]. First discovered in 1947, with subsequent studies reporting the isolation and characterization of additional Pmx derivatives from natural product sources [5,6,7,8], PmxB **1** and colistin **2** have been part of the clinical antibiotic repertoire for over 50 years, albeit approved for human use in an era with less stringent regulatory requirements compared to contemporary standards. However, toxicity issues, in particular nephrotoxicity [9,10], led to their gradual replacement with safer alternatives. The past decade has seen increasing application of ‘last-resort’ antibiotics due to the ominous rise of infections caused by extended-spectrum β-lactamase- (ESBL) and carbapenemase-producing strains of *Acinetobacter baumannii*, *Pseudomonas aeruginosa* and *Enterobacteriaceae*, with some strains now exhibiting multidrug resistance to practically all known antibiotics [11,12,13,14]. New treatments are urgently needed, but scientific [15] and economic [16] hurdles have slowed progression of the antibiotic clinical pipeline, particularly for Gram-negative therapies [17]. This has prompted a resurgence in the use of PmxB and colistin in spite of their toxicity, as well as renewed interest in the creation of improved analogues, with significant effort focused on optimising dosing strategies [18,19], developing a deeper understanding of structure-toxicity relationships [20,21], and investigating the clinical implications of the increasing prevalence of Pmx-resistance, which has come to the fore in recent years [22].

The polymyxins act initially by binding to lipid A, the membrane-anchoring component of lipopolysaccharide (LPS), which decorates the outer membrane of Gram-negative bacteria. The anionic nature of LPS facilitates electrostatic interaction with the pentacationic polymyxins. This initial interaction then leads to disruption of bacterial outer membrane permeability barrier through destabilisation of the LPS layer, and hydrophobic insertion of the fatty acyl chain of polymyxin into the lipid domain of lipid A. Subsequently, cytoplasmic membrane disruption and potential additional intracellular interactions lead to cell death [23]. In this context, an important structural feature of the polymyxins is the presence of multiple positively charged L-α-γ-diaminobutyric acid (Dab) side chains, which interact with the phosphate groups on lipid A [24]. However, this interaction alone is insufficient to kill bacteria, as polymyxins lacking a fatty acyl tail are poor antibiotics. For example, polymyxin B nonapeptide (PMBN), which contains an *N*-terminal Thr^2^-Dab^3^ dipeptide motif lacking a fatty acyl tail, is not antibacterial [25]. Thus, effective bacterial killing requires the concomitant interaction of LPS and the bacterial outer membrane with the Dab side chains and the lipid tail of polymyxin.

Over the years, several groups have attempted to develop next generation polymyxins with improved safety profiles. New compounds have been reported by Cubist [26], Pfizer [27], Cantab Anti-Infectives [28], MicuRx [29], University of Barcelona [30] and Northern Antibiotics [31]. Most strategies have focused on developing analogues designed to include the native lipodecapeptide structure of Pmx, as both the cyclic heptapeptide ring and the exocyclic lipotripeptide sequence are generally required for optimal antimicrobial activity. In contrast, the team led by Vaara at Northern Antibiotics have demonstrated antimicrobial potency for nonapeptide variants of Pmx in which the exocyclic fatty acyl-diaminobutyryl-threonyl-diaminobutyryl (FA-Dab^1^-Thr^2^-Dab^3^) linear tripeptide segment of Pmx was substituted with a truncated fatty acyl-dipeptide motif (FA-Thr^2^-XX^3^, where XX = d-Thr or d-Ser), exemplified by their compound NAB739 [31]. More drastic changes have been reported, including des-fatty acyl derivatives [32,33], but such compounds usually lack intrinsic antimicrobial activity, and instead act as membrane sensitizers that potentiate the activity of other antibiotics. PMBN is an archetypal example [25,34]. Spero Therapeutics have advanced a related analogue SPR-741 (formerly known as NAB741) into Phase I clinical trials; it maintains the PmxB heptapeptide ring, but incorporates an exocyclic fatty acyl-dipeptide motif Ac-Thr^2^-d-Ser^3^ [35].

We recently reported a systematic activity-toxicity study of PmxB, with a predominant focus on analogues maintaining the lipodecapeptide structure but with variations at every position [36]. As an extension of this study, we sought to examine the effect of *N*-terminal Dab-1 truncation, removing one of the positive charges purported to be associated with nephrotoxicity, leading to nonapeptide variants bearing an exocyclic lipodipeptide motif FA-aa^2^-aa^3^ (aa = amino acid) instead of the native tripeptide motif FA-Dab^1^-Thr^2^-Dab^3^ of Pmx (Figure 2). Herein we report the synthesis and biological evaluation of 36 unique Pmx nonapeptide analogues, alongside comparative biological data for 10 compounds that have been reported previously [33,36,37,38,39]. In the new nonapeptide series, we examined the effect of altering the fatty acid component, as well as the influence of variations at positions P2, P3 and P4 (numbering based on original Pmx decapeptide scaffold, with P4 the diamino acid residue involved in peptide cyclisation). Collectively the data has enabled a side-by-side comparison of truncated Pmx lipononapeptides versus their lipodecapeptide counterparts, in turn providing insight into the relative contribution of Dab-1 to antimicrobial potency. PmxB **1** and colistin **2** remained the most potent of all the compounds tested, but some nonapeptide analogues possessed similar potency to their decapeptide counterparts, contingent upon the fatty acyl component. Selected analogues also showed moderate activity toward a polymyxin-resistant clinical isolate of *P. aeruginosa* without appreciable cytotoxicity against human proximal tubular epithelial cells (HK-2).

## 2. Results and Discussion

### 2.1. Chemistry

A total of 36 compounds were synthesized in this study (ten of which have been reported previously [33,36,37,38,39]), with the structures presented in Table 1, Table 2 and Table 3 and Appendix A. All compounds possessed >95% purity, as determined by LCMS analysis using both ELSD and UV (210 nm) detection. The compounds were prepared by solid phase peptide synthesis (SPPS) (Scheme 1). Several SPPS strategies to construct the polymyxin scaffold have been reported, starting from the *C*-terminal Leu-7 [27] or Thr-10 [40,41] residue attached to the resin through the carboxyl terminus. Both strategies require a subsequent solution phase cyclisation step. On the other hand, on-resin cyclisation has been reported using Dab-9 as the anchoring point, which was attached to the resin via the N^γ^-amino group of the side chain [42]. In the present study, which utilizes an on-resin cyclisation strategy, the polymyxin scaffold was constructed by anchoring the side chain β-hydroxy group of the *C*-terminal Thr-10 residue onto a dihydropyran DHP HM resin, as previously reported (Scheme 1) [36]. The *C*-terminal carboxylic acid of Thr-10 and the N^γ^-amino group of Dab-4 were masked as allyl ester and carbamate protecting groups, respectively, allowing for orthogonal deprotection in the presence of the remaining Boc-protected Dab side chain amino groups en route to on-resin cyclisation. Thus, the synthesis was initiated using resin-bound Fmoc-L-Thr-CO_2_Allyl **3** (Scheme 1) [36]. The peptide sequence was constructed using SPPS with sequential Fmoc deprotection (30% piperidine in DMF) followed by Fmoc-amino acid coupling (HCTU, DIPEA in DMF), leading to the intermediate heptapeptide construct **4**. On-resin cyclisation was effected by in situ deprotection of both the C-terminus of Thr10 and the Dab-4 side chain amine using Pd(PPh_3_)_4_ and PhSiH_3_, generating **5**, which was then treated with DPPA and DIPEA in DMF overnight at room temperature to give the cyclised resin-bound intermediate **6**. The synthesis was then completed by removal of the Dab-4 α-amino Fmoc group followed by the sequential addition of the linear exocyclic tail residues Fmoc-L-Dab(Boc)-OH and Fmoc-L-Thr(^t^Bu)-OH to give the penultimate resin-bound precursor **7**. Polymyxin B nonapeptide (PMBN) was prepared from **7** by sequential Fmoc removal from Thr-2 followed by treatment with TFA/Et_3_SiH/H_2_O (95:1:4), which liberated the peptide from the resin with concomitant side chain deprotection. Analogues **3**–**10**, **12**–**14**, **16**–**28**, **30**–**44** and **46**–**47** were also prepared from **7** by sequential Fmoc removal from Thr-2 followed by acylation with the appropriate fatty acid, and cleavage/deprotection using TFA/Et_3_SiH/H_2_O (95:1:4). All compounds were purified by rp-HPLC and isolated as their TFA salts.

### 2.2. Biological Activity

All compounds synthesised in this study were assessed for their minimum inhibitory concentrations (MIC, mg/L) by broth microdilution assay against five antibiotic-sensitive and resistant ATCC reference strains covering the Gram-negative ESKAPE pathogens (*Escherichia coli*, *Klebsiella pneumoniae*, *A. baumannii*, and *P. aeruginosa*) (Table 1, Table 2 and Table 3). *Staphylococcus aureus* was also included as a representative Gram-positive bacterial strain. Antimicrobial profiling was also performed against a subset of polymyxin-resistant MDR clinical isolates of *K. pneumoniae*, *A. baumannii*, and *P. aeruginosa*, but most compounds were inactive (MIC > 32 mg/L), data not shown for *K. pneumoniae* and *A. baumannii* isolates. Polymyxin B **1**, colistin **2**, vancomycin and gentamicin were used as positive inhibitor comparator compounds. Compounds were counter-screened against human proximal tubular epithelial cells (HK-2), using LDH release as a general indicator of cellular toxicity [43,44,45].

Polymyxins without a fatty acyl tail lack antimicrobial potency, exemplified by **PMBN** [25], which was inactive against all strains except *P. aeruginosa* ATCC 27853 (MIC 2 mg/L) (Table 1). Interestingly, activity against *P. aeruginosa* ATCC 27853 was relatively insensitive to structural changes, with most compounds displaying MICs 1–4 mg/L despite a variety of structural modifications (Table 1, Table 2 and Table 3). **PMBN** potency was restored with incorporation of a C8 tail (octanoic acid, OA), as previously demonstrated by nonapeptide **8**, which possessed an exocyclic dipeptide motif OA-Thr^2^-Dab^3^ (MIC 1–2 mg/L for most strains) [36,37]. In comparison, **PmxB3** [33] incorporating the native exocyclic tripeptide motif FA-Dab^1^-Thr^2^-Dab^3^ of Pmx, was 2- to 4-fold more active than nonapeptide **8** [36,37] against most strains (Table 1). The observation that Pmx nonapeptides produced by truncation of L-Dab^1^ could retain activity when substituted with an appropriate fatty acid, exemplified by nonapeptide **8**, provided impetus to explore this phenomenon further.

In our previous study, we also reported decapeptide **16**, possessing a relatively polar 2-chorophenyl urea fatty acyl moiety [36]. This analogue displayed MICs of 1–2 mg/L across most of the tested Gram-negative strains (Table 1). In contrast, nonapeptide **15**, the truncated form of decapeptide **16**, revealed a more striking difference between the two analogues, with **15** possessing considerably reduced activity during pairwise comparison. This contrasts with the general retention of activity observed between compound **8** [36,37] and **PmxB3** [33]. This variation suggests a greater influence of L-Dab^1^ in the presence of a relatively polar fatty acid, and implies that judicious selection of the fatty acid in nonapeptides lacking the L-Dab^1^ may compensate for the reduced electrostatic component by additional hydrophobic interactions between the fatty acid and lipid A. This observation prompted further exploration of the fatty acyl component of analogue **8**. Additional alterations to the fatty acid tail were generally disfavoured (compare **10**–**15**), with **12** (FA = 4-hexylbenzoic acid) and **13** (FA = 6-phenylhexanoic acid) providing the best activities, albeit with increased cytotoxicity and some activity against *S. aureus* for **12** (MIC 8 mg/L). Data from this small SAR subset suggests that a simple *N*-alkyl fatty acyl chain is sufficient for good antimicrobial potency. The activity of **8** was abolished when the L-Thr^2^-L-Dab^3^ sequence was reversed (**9**, L-Dab^2^-L-Thr^3^).

The size of the macrocyclic ring was examined by substituting the diamino acid L-Dab^4^ in **8** with the homologs L-Orn^4^
**17** and L-Lys^4^
**18** (Table 1). Both modifications led to reduced activity, although the L-Orn^4^ variant was better tolerated, especially against *E. coli* and *P. aeruginosa* (MIC 2 mg/L against both). In contrast, reversal of the Dab^4^ stereochemistry to the D-configuration in **19** was highly detrimental (MIC > 32 mg/L for all strains). The data collectively suggests that optimal antimicrobial potency is highly dependent on the number of atoms forming the heptapeptide ring and the absolute configuration at P4.

The observations of Vaara [38] prompted an examination of the effect of reversing the stereochemistry of L-Dab^3^ in nonapeptide **8** to D-Dab^3^ in compound **21** (Table 2). Interestingly, the activity of **21** was somewhat comparable to **8**, highlighting the tolerance of P3 to stereochemical inversion. A similar result was observed when the same modification was applied to **PmxB3** leading to decapeptide **20**, which was made for comparison. Furthermore, the activities of both **20** and **21**, possessing exocyclic OA-Dab^1^-Thr^2^-D-Dab^3^ and OA-Thr^2^-D-Dab^3^ constructs, respectively, were notably comparable, suggesting the relative contribution of L-Dab^1^ toward potency was less important in this pairwise series compared to compounds **15** and **16** described earlier. Compound **21** was one of the few analogues examined that was active against a polymyxin-resistant strain of *P. aeruginosa* FADDI-PA070 (MIC 8 mg/L); cytotoxicity against HK-2 cells was promising (CC_50_ 289 µM). Encouraged by this result, variation of the fatty acid component of **21** was explored, providing compounds **22**–**27** (Table 2). Analogue **25**, containing 6-phenylhexanoic as the fatty acid component, possessed no observable cytotoxicity at the tested concentrations (CC50 > 300 µM) with consistent activity against the ATCC strains (MIC 0.5–4 mg/L), and moderate activity against polymyxin-resistant *P. aeruginosa* FADDI-PA070 (MIC 8 mg/L). On the other hand, more highly lipophilic fatty acids (e.g., **23, 24**) were poorly favoured, instead leading to increased cytotoxicity with accompanying Gram-positive activity (*S. aureus* MIC 4–16 mg/L), albeit with potent activity against the polymyxin-resistant *P. aeruginosa* FADDI-PA070 strain (MIC 2–4 mg/L). Compound **21** was also modified to include variants at P2, leading to analogues **28**–**33** (Table 2). Potency improvements were not observed when L-Thr-2 was replaced with other hydroxylated amino acids (Tyr, Ser, Hse), nor any of the other amino acids examined, suggesting that threonine may be optimal at position 2 in nonapeptides bearing a D-Dab^3^ substituent. Likewise, in the series **28**–**33**, inversion of the stereochemistry at position 3 (D-Dab^3^ to L-Dab^3^) was unproductive, with all compounds possessing MICs ≥ 16 mg/L (data not shown).

Finally, the promising activity observed for **21**, containing D-Dab^3^, prompted further exploration of P3 with both D- and L-amino acids, as well as glycine (Table 3). It was previously demonstrated [36] that Gly^3^ was poorly tolerated in nonapeptide **49**, in contrast to the potent activity observed for its decapeptide counterpart **50**, again emphasising the influence of L-Dab^1^ on potency between the nonapeptide and decapeptide series. Potency could be partially restored in **49** by substitution of the octanoic acid tail with alternative fatty acids (compounds **51** and **52**), albeit with variable activity across the Gram-negative panel (Table 3). Substitution of D-Dab^3^ in **21** with other basic amino acids (His, Lys, Orn, Dap) was generally well tolerated, even with inversion of stereochemistry, with **39** (L-Dap^3^) possessing the best activity (Table 3). Substitution with histidine was exceptional, with the D-isomer possessing much more potent activity compared to the L-isomer (compare **47** and **48**, Table 3). When D-Dab^3^ in **21** was modified to provide analogues **35** (L-Ser^3^) or **36** (d-Ser^3^), activity was significantly reduced (MIC 16–32 mg/L for most strains except *E. coli* where MIC = 4 mg/L), in contrast to data previously reported by Vaara (**35** = NAB743, **36** = NAB739) [31,38]. Of note, the decapeptide **34** [36] possessing an exocyclic OA-Dab^1^-Thr^2^-d-Ser^3^ construct, was significantly more potent than nonapeptide **36** lacking the L-Dab^1^ residue, again exemplifying the importance of L-Dab^1^ between nonapeptide and decapeptide variants. Activity across most strains was essentially abolished for analogues containing L-Hse^3^ (**37**), L-Asn^3^ (**38**), L-Cit^3^ (**44**) or L-Trp^3^ (**46**).

In the present study, we made a series of fatty *N*-acyl polymyxin nonapeptides, with alterations of the *N*-acyl component, and variation of the amino acids at positions P4, P3 and P2. Despite a lack of equipotency with polymyxin B **1** or colistin **2**, selected nonapeptide analogues possessed promising antimicrobial activity against the panel of five antibiotic-sensitive and resistant Gram-negative ATCC reference strains tested. Two analogues, **23** and **24**, were also active against a polymyxin-resistant strain of *P. aeruginosa*, but displayed increased cytotoxicity and activity against the Gram-positive *S. aureus*. Pairwise comparison of nonapeptide and decapeptide analogues (compare **8**, **9**; **15**, **16**; **20**, **21**; **34**, **36**; and **49**, **50**) revealed that the influence of the L-Dab^1^ residue on antimicrobial activity is contingent upon the fatty acyl component; activity was lost with more polar *N*-acyl components, but was retained with octanoic acid. Nonapeptides with stereochemical inversion at position 3 (i.e., D-Dab^3^ instead of L-Dab^3^) retained activity (e.g., **20** and **21**), as did analogues possessing alternative positively charged amino acids (e.g., **39**–**43**). On the other hand, activity was lost upon substitution of L-Dab^3^ with Gly^3^ in a nonapeptide framework, but could be restored in the decapeptide equivalent due to the presence of L-Dab^1^ (compare **49** and **50**). A similar trend was evident upon substitution of L-Dab^3^ with a neutral d-Ser^3^ residue (compare **34** and **36**). Collectively the present study demonstrates that polymyxin B nonapeptides may find utility in the design of improved polymxin analogues to fight antibiotic resistant infections.

## 3. Materials and Methods

### 3.1. Synthesis

Experimental procedures are described in the Appendix A. All chemicals were obtained from commercial suppliers and used without further purification. LC-MS analyses were conducted using Agilent Technologies 1200 Series Instrument with a G1316A variable wavelength detector set at λ = 210 nm, 1200 Series ELSD, 6110 quadrupole ESI-MS, using an Agilent Eclipse XDB-Phenyl column (3 × 100 mm, 3.5 μm particle size, flow rate 1 mL/min, the mobile phases 0.05% formic acid in water and 0.05% formic acid in acetonitrile) (Agilent Technologies, Melbourne, Australia). Compound purification was performed using an Agilent 1260 Infinity Preparative HPLC with a G1365D multiple wavelength detector set at λ = 210 nm coupled to an Agilent Eclipse XDB-Phenyl column (21.2 × 100 mm, 5 μm particle size). Identities of final products were confirmed by high resolution mass spectrometry (HRMS), performed on a Bruker Micro TOF mass spectrometer using (+)-ESI calibrated to sodium formate (Bruker Daltonics, Melbourne, Australia). Final purity of more than 95% for all compounds was confirmed by LC-MS analysis using both ELSD and UV (210 nm) detection.

### 3.2. Minimum Inhibitory Concentration (MIC) Determination by Broth Microdilution Assay

Bacteria were either obtained from American Type Culture Collection (ATCC; Manassas, VA, USA) or independent academic clinical isolate collections, as listed in Appendix A. Bacteria were cultured in nutrient broth (NB; Bacto Laboratories, catalog No. 234000) or Mueller Hinton broth (MHB; Bacto Laboratories, catalog No. 211443) at 37 °C overnight with shaking (∼180 rpm). A sample of each culture was then diluted 50-fold in fresh MHB and incubated at 37 °C for 1.5–3 h with shaking (∼180 rpm). Compound stock solutions were prepared as 0.64 or 2.56 mg/mL in water. The compounds, at twice the final desired concentration, were serially diluted 2-fold across the wells of 96-well plates (Polystyrene, Corning, catalogue No. 3370). Mid log phase bacterial cultures (after 1.5–3 h incubation) were diluted to 1 × 10^6^ colony forming units (CFU)/mL, and 50 μL was added to each well giving a final compound concentration range of 32 mg/L to 0.015 mg/L and a final cell density of 5 × 10^5^ CFU/mL. MICs were determined visually after 18 h of incubation at 37 °C, with the MIC defined as the lowest compound concentration at which no bacterial growth was visible.

### 3.3. Cytotoxicity (Lactate Dehydragenase (LDH) Assay)

Cytotoxicity to human kidney proximal tubular epithelial cell line, HK-2 (ATCC CRL-2190, sourced from ATCC; Manassas, VA, USA) was determined using the LDH assay as previously described [43,44,45]. In brief, HK-2 cells were seeded as 2000 cells/well in black-walled clear bottom 384-well tissue culture treated plates (Corning, catalogue No. 3712) in DMEM/F12 medium (Gibco^®^ 10565-042) containing 10% of Fetal Bovine Serum (FBS, Gibco^®^ 10099-141) and incubated for 24 h at 37 °C, 5% CO_2_. Compounds were then added into each well with a concentration series from 300 µM to 2.3 µM in 2-fold dilutions. Colistin and polymyxin B were used as controls and tested at a final concentration range of 1 mM to 7.8 µM. The cells were incubated with the compounds for 24 h at 37 °C, 5% CO_2_. After the incubation, 5 µL of culture medium was added to 45 µL of LDH assay buffer (Biovision, K313-500) and incubated for 30 min at room temperature. The absorbance (ABS) was then read at 450 nm using a Polar Star Omega plate reader. The data was analysed by Prism 6 software (GraphPad Software, La Jolla, CA, USA). Results were calculated using the following equation: cytotoxicity %= (ABS_samples_ − ABS_untreated_/ABS_1%Triton X-100_ − ABS_untreated_) × 100.

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
