# Peer review of "Structure-Function Studies of Polymyxin B Lipononapeptides"

_molecules, 2019, doi:10.3390/molecules24030553_

Round 1
Reviewer 1 Report
This work represents an interesting contribution to the polymixin related research and merits to be publish in Molecules.
The only concern of this referee is that the manuscript is difficult to follow. I would suggest to introduce new columns in the tables indicating clearly the nature of the aa P1, P2, P3. Without this, it is very difficult to understand
Author Response
Additional columns have been added to the Tables as suggested.
Reviewer 2 Report
The authors present a SAR study of Polymyxin B lipononapeptides with a focus on the exocyclic motif (FA-Dab1-Thr2-Dab3) and on the number of atoms in the heptameric ring (position P4). Particularly, truncated analogs with Dab1 residue removed and various mutants at P3 were extensively studied. These results provide a better understanding of how to develop new antibiotics based on the known polymyxin.
I recommend publishing the manuscript in its current state after a small correction. Namely, in the scheme that belongs to Tables 2 structures for peptides 8-19 are shown instead of 20-33. This must be corrected.
Author Response
Table 2 Scheme has been corrected.